# Is the Effect of the COVID-19 Vaccine on Heart Rate Variability Permanent?

**DOI:** 10.3390/medicina59050852

**Published:** 2023-04-28

**Authors:** Murat Kerkutluoglu, Hakan Gunes, Ufuk Iyigun, Musa Dagli, Adem Doganer

**Affiliations:** 1Department of Cardiology, Faculty of Medicine, Kahramanmaras Sutcu Imam University, Kahramanmaras 46050, Turkey; drhakangunes83@gmail.com (H.G.);; 2Department of Cardiology, Hatay Training and Research Hospital, Hatay 3100, Turkey; 3Department of Biostatistics and Medical Informatics, Faculty of Medicine, Kahramanmaras Sutcu Imam University, Kahramanmaras 46050, Turkey

**Keywords:** COVID-19, vaccine, Pfizer-BioNTech, heart rate variability, autonomic dysfunctions

## Abstract

*Background and Objectives:* The risk of autonomic dysfunction with COVID-19 vaccines used worldwide in the COVID-19 pandemic remains a topic of debate. Heart rate variability has a number of parameters that can be used to assess autonomic nervous system dynamics. The aim of this study was to investigate the effect of a COVID-19 vaccine (Pfizer-BioNTech) on heart rate variability and autonomic nervous system parameters, and the duration of the effect. *Materials and Methods:* A total of 75 healthy individuals who visited an outpatient clinic to receive the COVID-19 vaccination were included in this prospective observational study. Heart rate variability parameters were measured before vaccination and on days 2 and 10 after vaccination. SDNN, rMSSD and pNN50 values were evaluated for time series analyses, and LF, HF, and LF/HV values for frequency-dependent analyses. *Results:* The SDNN and rMSDD values declined significantly on day 2 after vaccination, while the pNN50 and LF/HF values increased significantly on day 10. The values at pre-vaccination and at day 10 were comparable. The pNN50 and LF/HF values declined significantly on day 2 and increased significantly on day 10. The values at pre-vaccination and at day 10 were comparable. *Conclusions:* This study showed that the decline in HRV observed with COVID-19 vaccination was temporary, and that the Pfizer-BioNTech COVID-19 vaccination did not cause permanent autonomic dysfunction.

## 1. Introduction

The COVID-19 pandemic has become a global challenge, infecting over 757 million people and causing the death of approximately 6.8 million patients around the world [1]. Vaccination is essential to preventing a pandemic. The first vaccine that was given fast-track approval by the U.S. Food and Drug Administration to prevent COVID-19 infection was a BNT162b2 messenger RNA (mRNA) COVID-19 vaccine (Pfizer-BioNTech COVID vaccine), which encoded the spike protein of SARS [2,3]. Following preliminary studies that showed a neutralizing antibody response with this mRNA vaccine, a phase 3 randomized clinical trial demonstrated that the Pfizer-BioNTech vaccine was safe and had 95% efficacy vs. a placebo in reducing the risk of contracting COVID-19 [3,4,5].

Heart rate variability (HRV) is thought to be an accurate and non-invasive way to measure the sympathetic and parasympathetic parts of the autonomic nervous system. A reduction in HRV, a strong indicator of autonomic function, is closely associated with mortality. It is also known that both acute and chronic inflammation are likewise closely related to autonomic dysfunction [6,7,8,9,10,11]. Moreover, it has been shown that autonomic dysfunction is involved in subclinical inflammatory processes [12]. A limited number of studies that investigated the effects of an inflammatory response to vaccination have reported autonomic tone changes secondary to the low-grade inflammation [13]. The effect of mRNA vaccines, developed using state-of-the-art technologies, on autonomic function, is not fully known.

COVID-19 immunization may be associated with autonomic nervous system (ANS) dysfunction, which has been described in certain case reports [14,15,16,17,18,19]. Therefore, it is unknown if COVID-19 immunization actually causes autonomic dysfunction; however, anecdotal occurrences are likely to heighten public fear and, thus, vaccine reluctance. Previously reported associations between other vaccines and autonomic dysfunction, including human papillomavirus (HPV) and influenza A, may be partially explained by the systemic inflammatory response, and cases of the multisystem inflammatory syndrome (MIS) associated with COVID-19 vaccination have been reported recently [20,21,22].

In this investigation, we looked at the potential influence of COVID-19 vaccination on human HRV-related parameters, notably vaccination safety, in order to acquire information for the distribution of evidence-based support for COVID-19 vaccination.

## 2. Materials and Methods

### 2.1. Study Population

The study was conducted between October 2021 and December 2021 on individuals who visited a vaccination outpatient clinic at Hatay Teaching and Research Hospital to receive the vaccination. There were 74 healthy subjects who visited the vaccination outpatient clinic and agreed to participate in the study. Patients with a history of COVID-19 infection; active smokers; patients with hypertension and/or diabetes mellitus; patients using beta blockers or calcium channel blockers; patients with moderate to severe valvular heart disease; patients with a prosthetic heart valve; or patients with coronary artery disease, left ventricular dysfunction, atrial fibrillation, frequent atrial or ventricular extra beats, chronic obstructive pulmonary disease, asthma, obstructive sleep apnea, a body mass index of over 30 kg/m^2^, renal insufficiency, cerebrovascular disease, thyroid disease, chronic liver disease or inflammatory or autoimmune disorders, were excluded. Care was taken to ensure that female subjects were not in their active menstrual cycle. A blinded cardiologist performed an echocardiographic examination of the patients. Patients with abnormal echocardiographic findings were excluded. All included patients received COVID-19 testing prior to vaccination. The HRV values of the enrolled subjects were analyzed prior to vaccination and at days 2 and 10 after vaccination.

### 2.2. HRV Analysis

Electrocardiography with standard 12 derivations (PC-ECG 1200, Norav Medical Ltd., Yokneam, Israel) was performed on the study subjects. Fixed 10-minute ECG segments were recorded for each subject while they were breathing normally at rest in the supine position in an unsupervised setting. Heart Rate Variability Software (version 4.2.0, Norav Medical Ltd., Yokneam, Israel) was used to analyze heart rate variability. Both time series and frequency-dependent analyses were performed.

The following parameters were used for time series analyses: SDNN, ms: standard deviation of intervals between normal heartbeats (R-R);rMSSD, ms: root mean square of successive differences between normal heartbeats;pNN50, %: percentage of instances of a difference of more than 50 milliseconds between successive heartbeats.The following parameters were used for frequency-dependent analyses:HF: high frequency, covering a frequency range of 0.16–0.4 Hz, indicating parasympathetic activity;LF: low frequency, covering a frequency range of 0.04–0.15 Hz, indicating sympathetic activity;LF/HF: a ratio indicating sympathovagal activity, where a higher value means increased sympathetic activity;TP: total power, signifying a variation in time intervals between all heart beats.

We determined the power spectrum within the predefined frequency bands of interest: very-low frequency (<0.04 Hz), low frequency (0.04–0.15 Hz) and high frequency (0.14–0.4 Hz) components, displayed in absolute values of power (milliseconds squared).

### 2.3. Statistical Analysis

The Kolmogorov–Smirnov test was used for analyzing the conformity of quantitative variables to a normal distribution. The difference between repeated measurements of conformant variables was analyzed using repeated measures ANOVA. Bonferroni post hoc analysis was used for comparing dual measures. The Friedman test was used for evaluating the differences between re-measurements of outliers. Dunn post hoc analysis was used for comparing dual measures. The statistical parameters were expressed as mean ± standard deviation and median (first quartile (25th percentile)–third quartile (75th percentile). The statistical significance was set to *p* < 0.05. SPSS version 22 (IBM SPSS for Windows version 22, IBM Corporation, Armonk, NY, USA) and R 3.3.2 software were used for evaluating the data. 

## 3. Results

Of the subjects, 59% were male and 41% were female. The mean age of the study population was 45 ± 9 years. Based on the measured HR, SDNN and rMSSD values of the subjects, the HR values increased significantly on day 2 after vaccination (67.90 ± 8.50 vs. 87.10 ± 10.40, *p* < 0.001) and declined significantly at day 10 compared to day 2 (87.10 ± 10.40 vs. 72.60 ± 8.60, *p* < 0.001); the pre-vaccination values and the values at day 10 after vaccination were comparable (Table 1). The SDNN and rMSDD values declined significantly on day 2 after vaccination (61.60 ± 11.09 vs. 32.35 ± 8.52, *p* < 0.001 and 34.55 ± 7.62 vs. 19.47 ± 6.90, *p* < 0.001, respectively), and increased significantly at day 10 compared to day 2 (32.35 ± 8.52 vs. 60.37 ± 10.98, *p* < 0.001 and 19.47 ± 6.90 vs. 33.35 ± 7.64, *p* < 0.001, respectively); the pre-vaccination values and the values at day 10 were comparable (Figure 1a).

The LF values decreased significantly on day 2 after vaccination [151.42 (42.15–244.23) vs. 63.10 (27.20–174.30), *p* < 0.001], and the values increased significantly at day 10 after vaccination compared to day 2 [63.10 (27.20–174.30) vs. 161.94 (38.50–247.46), *p* < 0.001]; the pre-vaccination values and the values at day 10 were comparable (Table 1).

The total power and HF values declined significantly on day 2 after vaccination [282.41 (169.64–587.28) vs. 264.87 (139.75–549.51), *p* < 0.001 and 130.04 (51.64–252.54) vs. 88.24 (22.92–223.43), *p* < 0.001, respectively] and the values increased significantly at day 10 after vaccination compared to day 2 [264.87 (139.75–549.51) vs. 272.11 (166.87–585.72), *p* < 0.001 and 88.24 (22.92–223.43) vs. 130.49 (49.96–256.65), *p* < 0.001, respectively] (Table 1). The pre-vaccination values and the values on day 10 were comparable (Figure 1b).

The VLF values were comparable at pre-vaccination, at day 2 after vaccination and at day 10 after vaccination [199.50 (110.04–359.98) vs. 141.95 (76.85–217.82) vs. 198.84 (110.73–360.34), *p* = 0.420] (Table 1), (Figure 1b).

The pNN50 and LF/HF values decreased significantly on day 2 after vaccination [0.51 (0.23–1.52) vs. 0.27 (0.11–0.60), *p* < 0.001 and 0.99 (0.53–1.95) vs. 0.74 (0.39–1.13), *p* < 0.001, respectively], and the values increased significantly at day 10 after vaccination compared to day 2 [0.27 (0.11–0.60) vs. 0.52 (0.25–1.48), *p* < 0.001 and 0.74 (0.39–1.13) vs. 0.96 (0.54–1.99), *p* < 0.001, respectively]; the pre-vaccination values and the values at day 10 were comparable (Table 1, Figure 1c).

## 4. Discussion

A main finding of this study was that the SDNN, RMSSD and pNN50 values used for the time series analyses, and the LF, HF, LF/HF and TP values used for frequency-dependent analyses of heart rate variability, decreased at day 2 after vaccination and increased at day 10 over day 2, returning to comparable levels with those at pre-vaccination. It is possible to distinguish between the parasympathetic and sympathetic effects using time series and frequency-based analyses of heart rate variability. The time series parameter SDNN indicates the overall balance state of the autonomic nervous system, and pNN and RMSSD values predominantly indicate parasympathetic activity [23]. HF is a dominant indicator of parasympathetic activity. LF is associated with baroreceptor control of sympathetic activity, and may also be affected by parasympathetic activity. Therefore, LF is affected by both the sympathetic and parasympathetic systems. The underlying mechanism of VLF is not fully understood. The LF/HF ratio is believed to be typically indicative of sympathovagal balance [24].

It has been shown that reduced HRV, an indicator of autonomic dysfunction, is associated with a poor prognosis in infectious diseases. It is believed that a reduction in HRV may be linked to the inflammatory and immune responses in infectious diseases [25,26,27]. The inflammatory response, which also occurs after vaccination, may initially cause autonomic dysfunction. It has been shown that the reduction in HRV that occurs after influenza vaccination is closely related to elevated CRP levels [21]. A previous study demonstrated that increased sympathetic activity of the autonomic nervous system is an immunomodulatory effect of reduced parasympathetic activity [28]. It is believed that increased sympathetic activity may be an indicator of reduced parasympathetic activity; in other words, a reduction in HRV may be an indicator of a protective inflammatory response or an elevated immune response [29]. Therefore, it was assumed that the initial decrease in HRV was a manifestation of a stronger immune response to the COVID-19 vaccination [30,31,32]. Moreover, in our study, both the time series and frequency-based analyses showed that the reduction in HRV is associated with elevated sympathetic activity and diminished parasympathetic activity. In view of this information, we consider that the decrease in HRV and increased activity of the sympathetic nervous system, which occurred on day 2 of our study, served to increase the immune response. Notably, at day 10, the HRV values of subjects returned to comparable levels with those measured at pre-vaccination, which may be indicative of a completed inflammatory or antibody response. Similar to our findings, Hajduczok et al. reported that HRV values, measured using wearable devices, declined on day 2 after vaccination and, in the days that followed, returned to comparable levels with those measured at pre-vaccination [33]. Presbay et al. demonstrated that cardiovascular measures varied from baseline values on the first night after COVID-19 vaccination, but that the worsening was temporary, and the values reverted to baseline levels within four nights of vaccination [34]. Gepner et al. found that within three days after vaccination, all physiological measurements returned to their pre-vaccination levels [35]. In our study, as in these studies, we found that the physiological response to COVID-19 vaccination was temporary and reversible, with most measures returning to pre-vaccination levels within a few days. This information may help alleviate concerns about the potential long-term effects of the vaccine on cardiovascular and autonomic nervous system function.

## 5. Conclusions

This study showed that temporary changes were observed in HRV after COVID-19 vaccination. Generally, it was demonstrated that this brief change returned to normal on day 10 after vaccination, and that vaccination did not cause permanent autonomic dysfunction. This study supports the safety of the COVID-19 vaccination. This study was conducted in a small cohort without a control group, and it was not designed as a randomized, controlled study. Since healthy individuals were included in the study, biomarkers such as interleukins and CRP, indicators of the inflammatory process, were not considered. Another limitation of this study is that subjects could not be tested for antibodies, which may be an indicator of the immune response. 

## Figures and Tables

**Figure 1 medicina-59-00852-f001:**
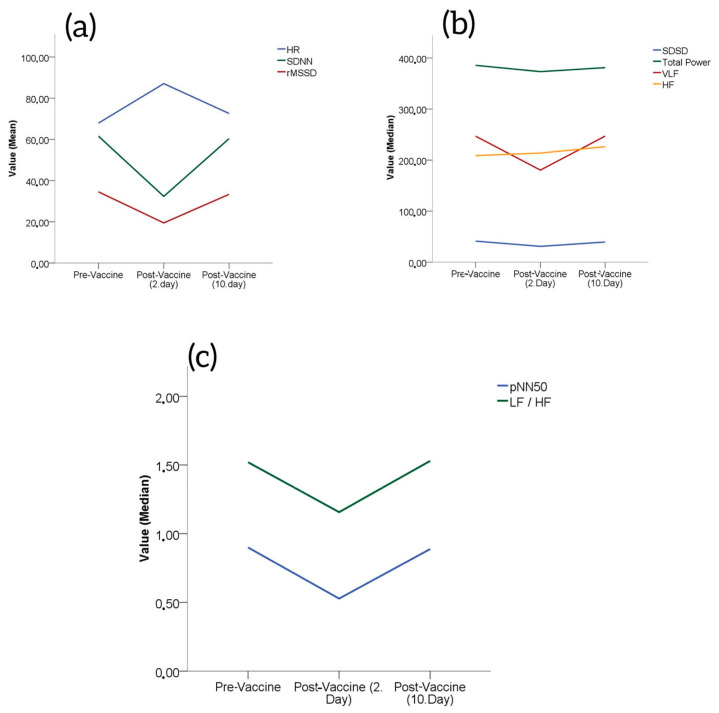
(**a**) Changes in HR, SDNN and rMSSD from baseline after Pfizer/BioNTech vaccinations. (**b**) Changes in total power, VLF and HF from baseline after Pfizer/BioNTech vaccinations. (**c**) Changes in pNN50 and LF/HF from baseline after Pfizer/BioNTech vaccinations.

**Table 1 medicina-59-00852-t001:** Comparison of Heart Rate Variability parameters.

	Pre-Vaccination	2nd Day of Vaccination	10th Day of Vaccination	*p*
HR ^y^, Mean ± SD	67.90 ± 8.50 ^b^	87.10 ± 10.40 ^a,c^	72.60 ± 8.60 ^b^	*p* < 0.001 *
SDNN ^y^, Mean ± SD	61.60 ± 11.09 ^b^	32.35 ± 8.52 ^a,c^	60.37 ± 10.98 ^b^	*p* < 0.001 *
rMSSD ^y^, Mean ± SD	34.55 ± 7.62 ^b^	19.47 ± 6.90 ^a,c^	33.35 ± 7.64 ^b^	*p* < 0.001 *
TotalPower ^z^, Median (Q1–Q3)	282.41 (169.64–587.28) ^b^	264.87 (139.75–549.51) ^a,c^	272.11 (166.87–585.72) ^b^	*p* < 0.001 *
VLF ^z^, Median (Q1–Q3)	199.50 (110.04–359.98)	141.95 (76.85–217.82)	198.84 (110.73–360.34)	0.420
LF ^z^, Median (Q1–Q3)	151.42 (42.15–244.23) ^b^	63.10 (27.20–174.30) ^a,c^	161.94 (38.50–247.46) ^b^	*p* < 0.001 *
HF ^z^, Median (Q1–Q3)	130.04 (51.64–252.54) ^b^	88.24 (22.92–223.43) ^a,c^	130.49 (49.96–256.65) ^b^	*p* < 0.001 *
pNN50 ^z^, Median (Q1–Q3)	0.51 (0.23–1.52) ^b^	0.27 (0.11–0.60) ^a,c^	0.52 (0.25–1.48) ^b^	*p* < 0.001 *
LF/HF ^z^, Median (Q1–Q3)	0.99 (0.53–1.95) ^b^	0.74 (0.39–1.13) ^a,c^	0.96 (0.54–1.99) ^b^	*p* < 0.001 *

^y^ Repeated measures ANOVA: post hoc: LSD test; a: 0.05; ^z^ Friedman test; post hoc: Dunn test; a: 0.05; * difference between groups is significant; ^a^ difference with pre-vaccine measurement values is significant; ^b^ difference between the 2nd day of head measurements is significant; ^c^ difference is significant by measurements at 10 days of vaccination.

## Data Availability

Data are available upon request from the corresponding authors.

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
