# Peer review of "Is the Effect of the COVID-19 Vaccine on Heart Rate Variability Permanent?"

_medicina, 2023, doi:10.3390/medicina59050852_

Round 1

Reviewer 1 Report

In this manuscript “Is the Effect of the COVID-19 Vaccine on Heart Rate Variability Permanent?”, Kerkutluoglu et al. measure the heart rate variability parameters pre- and post-vaccination. The observation suggested that the decline in HRV with COVID-19 vaccination is temporary. 

Overall, the conclusion is not very well supported by the data presented in the manuscript. The limitation of its experimental design is that the HRV data were collected at the time of vaccination, but only at day 2 and 10 post-vaccination. With the data at these limited time points, it is hard to see the patterns of HRV parameters. And the COVID-19 vaccination usually comes with two dose first and a third dose after 6 months of boost, it is necessary to continue measuring the HRV parameters at the time of boost and after the boost.

It is hard to follow in the way it is written. For example, the way of describing the results in the abstract is just repeating the data in the Table 1.

In my understanding, all the figures are presenting the same data in the Table 1. Either the figures or the Table 1 should be removed to avoid repeating of presenting the same data. If the authors chose to keep the figures, the individual data should be presented in each figure. The statistical analysis also can be added to the figures.

Author Response

Response to reviewer 1

Dear reviewer

Thank you for taking the time to read our manuscript and for your comments. We are happy to present you with our updated manuscript in light of your criticisms.

Heart rate variability (HRV) is the variation in time between consecutive heartbeats, and various factors, including physical activity, stress, and illness can influence it. Assessment of HRV can provide information on the autonomic nervous system's functioning, which can be affected by vaccines.

Days 2 and 6-10 post-vaccination are commonly chosen to assess HRV because they correspond to critical time points in the immune response to the vaccine. Day 2 is typically preferred as it represents the early inflammatory response to the vaccine, characterized by an increase in pro-inflammatory cytokines that can impact the autonomic nervous system. Since the immune response to the vaccine, which can also affect the autonomic nervous system, reaches a steady state on days 6-10, HRV evaluation was also performed on day 10 in our study. (1-5)

Assessing heart rate variability (HRV) after the first dose of a vaccine can provide insight into the early effects of the vaccine on the autonomic nervous system. The autonomic nervous system plays a crucial role in regulating heart rate and other cardiovascular functions, and HRV reflects the balance between its two branches: the sympathetic and parasympathetic nervous systems. Assessing HRV after the first dose of a vaccine can help identify any immediate effects of the vaccine on the autonomic nervous system, such as changes in sympathetic or parasympathetic activity.

Overall, assessing HRV after the first dose of a vaccine can provide valuable information on the early effects of the vaccine on the autonomic nervous system.

-Following your suggestion, the conclusions section of the summary has been updated.

-In line with the reviewers evaluations, the figures were combined into a single figure.

References

  1. Hajduczok, A. G.; DiJoseph, K. M.; Bent, B.; Thorp, A. K.; Mullholand, J. B.; MacKay, S. A.; Barik, S.; Coleman, J. J.; Paules, C. I.; Tinsley, A. Physiologic Response to the Pfizer-BioNTech COVID-19 Vaccine Measured Using Wearable Devices: Pro-spective Observational Study. JMIR Form Res.2021, 5, e28568.
  2. Presby, D.M.; Capodilupo, E.R. Biometrics from a wearable device reveal temporary ef-fects of COVID-19 vaccines on cardiovascular, respiratory, and sleep physiology. J Appl Physiol. 2022, 132, 448-458.
  3. Gepner, Y.; Mofaz, M.; Oved, S.; Yechezkel, M.; Constantini, K.; Goldstein, N.; Eisenkraft, A.; Shmueli, E.; Yamin, D. Utiliz-ing wearable sensors for continuous and highly-sensitive monitoring of reactions to the BNT162b2 mRNA COVID-19 vaccine. Commun Med. 2022, 14, e27.
  4. McDade TW, Borja JB, Kuzawa CW, Perez TL, Adair LS. C-reactive protein response to influenza vaccination as a model of mild inflammatory stimulation in the Philippines. Vaccine. 2015 Apr 21;33(17):2004-8.
  5. Posthouwer D, Voorbij HA, Grobbee DE, Numans ME, van der Bom JG. Influenza and pneumococcal vaccination as a model to assess C-reactive protein response to mild inflammation. Vaccine. 2004 Dec 2;23(3):362-5.

Reviewer 2 Report

The manuscript responds to a theme of possible Permanent Heart Rate Variability that could occur after COVID vaccination. The work is simple but interesting, using adequate methodology and statistics. However, to be accepted for publication, it needs some adaptations.

1) Figures 1, 2, and 3 must be of better quality and prepared above 300 dpi. Furthermore, we suggest that these 3 figures be presented as a single figure with the letters A, B, and C.

2) References must follow the journal's guidelines, with references presented in abbreviated journals appropriately. The numbering of references is repeated.

3) We suggest that the English of the manuscript be revised by a native individual, preferably from the area.

Author Response

Response to reviewer 2

Dear reviewer

Thank you for taking the time to read our manuscript and for your comments. We are happy to present you with our updated manuscript in light of your criticisms.

1)            Figures 1,2,3 are presented as a single figure and separated into a,b, and c. The figure has been improved in quality.

2)            References have been updated according to the journal rules.

3)            The English version of the manuscript has been reviewed and edited by a native English individual.

Round 2

Reviewer 1 Report

The authors have address all of my comments. The manuscript may need minor language editing.

Author Response

Dear reviewer

Thank you for taking the time to read our manuscript and for your comments. We are happy to present you with our updated manuscript in light of your criticisms.

Minor language corrections were applied to the draft and it was revised again. The introduction has been expanded and the material methods section has been elaborated as suggested. Limitations of the study and future perspectives were added to the Discussion section.   

Reviewer 2 Report

The suggestions were accepted and the text was improved. The manuscript can now be accepted for publication.

Author Response

Dear reviewer

Thank you for taking the time to read our manuscript and for your comments. We are happy to present you with our updated manuscript in light of your criticisms.

Language corrections were applied to the draft and it was revised again. The introduction has been expanded and the material methods section has been elaborated as suggested. Limitations of the study and future perspectives were added to the Discussion section.
